# The Effect of Stress Hormones, Ultraviolet C, and Stilbene Precursors on Expression of Calcineurin B-like Protein (*CBL*) and CBL-Interacting Protein Kinase (*CIPK*) Genes in Cell Cultures and Leaves of *Vitis amurensis* Rupr

**DOI:** 10.3390/plants12071562

**Published:** 2023-04-05

**Authors:** Konstantin V. Kiselev, Olga A. Aleynova, Zlata V. Ogneva, Andrey R. Suprun, Alexey A. Ananev, Nikolay N. Nityagovsky, Alina A. Dneprovskaya, Alina A. Beresh, Alexandra S. Dubrovina

**Affiliations:** 1Laboratory of Biotechnology, Federal Scientific Center of the East Asia Terrestrial Biodiversity, Far Eastern Branch of the Russian Academy of Sciences, Vladivostok 690022, Russia; 2Department of Biotechnology, The School of Natural Sciences, Far Eastern Federal University, Vladivostok 690090, Russia

**Keywords:** calcium, CBL, CIPK, plant secondary metabolites, stilbenes, grapevine, methyl jasmonate, piceid, salicylic acid, UV-C

## Abstract

Calcium serves as a crucial messenger in plant stress adaptation and developmental processes. Plants encode several multigene families of calcium sensor proteins with diverse functions in plant growth and stress responses. Several studies indicated that some calcium sensors may be involved in the regulation of secondary metabolite production in plant cells. The present study aimed to investigate expression of calcineurin B-like proteins (*CBL*) and CBL-interacting protein kinase (*CIPK*) in response to conditions inducting biosynthesis of stilbenes in grapevine. We investigated *CBL* and *CIPK* gene expression in wild-growing grapevine *Vitis amurensis* Rupr., known as a rich stilbene source, in response to the application of stilbene biosynthesis-inducing conditions, including application of stress hormones (salicylic acid or SA, methyl jasmonate or MeJA), phenolic precursors (*p*-coumaric acids or CA), and ultraviolet irradiation (UV-C). The influence of these effectors on the levels of 13 *VaCBL* and 27 *VaCIPK* mRNA transcripts as well as on stilbene production was analyzed by quantitative real-time RT-PCR in the leaves and cell cultures of *V. amurensis*. The data revealed that *VaCBL4-1* expression considerably increased after UV-C treatment in both grapevine cell cultures and leaves. The expression of *VaCIPK31*, *41-1*, and *41-2* also increased, but this increase was mostly detected in cell cultures of *V. amurensis*. At the same time, expression of most *VaCBL* and *VaCIPK* genes was markedly down-regulated both in leaves and cell cultures of *V. amurensis*, which may indicate that the *CBLs* and *CIPKs* are involved in negative regulation of stilbene accumulation (*VaCBL8*, *10a-2*, *10a-4*, *11*, *12*, *VaCIPK3*, *9-1*, *9-2*, *12*, *21-1*, *21-2*, *33*, *34*, *35*, *36*, *37*, *39*, *40*, *41-3*, *41-4*). The results obtained provide new information of CBL and CIPK implication in the regulation of plant secondary metabolism in response to stress hormones, metabolite precursors, and UV-C irradiation.

## 1. Introduction

Calcium serves as a crucial messenger in plant responses to environmental stresses and many developmental processes, including pollen tube elongation, cell polarity after fertilization, cell division, seed germination, apoptosis, phytohormone responses, senescence, abiotic stress tolerance, pathogens, etc. [1,2,3].

The plant calcium sensor proteins include calmodulins (CaMs), calmodulin-like proteins (CML), calcium-dependent protein kinases (CDPKs), calcineurin B-like proteins (CBLs), calcineurin B-like proteins (CBL), and CBL-interacting protein kinases (CIPK) [4,5,6]. There are scarce data on the functions and regulation of the CBL–CIPK calcium signaling network. A novel type of calcium sensor, termed as CBLs, was identified in plant cells 20 years ago [7]. CBLs specifically target a family of plant-specific CIPK. To decode a calcium signal, CBL binds Ca^2+^ and interacts with CIPK, leading to activation of the kinase. Then, the CBL–CIPK complex phosphorylates downstream target proteins and changes their biological activities [8]. Data from the current literature report on the functions of the plant CBL–CIPK network in mineral uptake, stomatal movement, pH regulation, maintenance of Na+/K+ homeostasis, etc. [8].

Ishitani et al. [9] demonstrated that the CBL4–CIPK24 complex of *Arabidopsis thaliana* migrated to the plasma membrane where it activated the Na^+^/H^+^ antiporter (SOS1) located on the plasma membrane and the vacuolar H^+^-ATPase, resulting in enhanced plant salt tolerance. Other CBL–CIPK complexes have been studied in Arabidopsis, such as CBL1/CBL9–CIPK23 that was observed to localize to the plasma membrane and regulate K^+^ in roots and stomatal guard cells [10]. Overexpression of *OsCBL8* and *OsCIPK15* in rice enhanced plant salt tolerance [11]. *OsCIPK31* mutants exhibited hypersensitive phenotypes to abscisic acid (ABA), salt, mannitol, and glucose. Compared with wild-type rice plants, *OsCIPK31* mutants exhibited retarded germination and slow seedling growth [12]. Moreover, *TaCIPK23* has been shown to participate in wheat cell signaling in response to ABA and drought by mediating the crosstalk between ABA-induced signaling and other pathways associated with adaptation to drought [13]. In potatoes, *StCIPK10* has been shown to enhance both cellular scavenging of reactive oxygen species and cellular modulation of corresponding osmoregulatory substances to strengthen plant drought and osmotic stress tolerance [14].

A number of articles have shown that plant calcium sensor proteins, such as CDPKs and CMLs, are involved in the plant pathogen response [15,16] and plant defense tools such as secondary metabolite production [17]. It is proposed that CDPKs are involved in stress hormone- and light-mediated activation of plant defense reactions and secondary metabolite production in response to biotic and abiotic cues [16]. Considering CBL and CIPK, there is scarce information on their involvement in plant defense reactions. It is known that overexpression of *StCBL4* and *StCIPK2* individually and synergistically increased the tolerance of potato plants to soilborne plant pathogen fungi *Rhizoctonia solani* in *Nicotiana benthamiana* [18].

Knockdown of the *TaCIPK14* gene in wheat significantly increased wheat resistance to the pathogenic fungus *Puccinia striiformis* (Pst), whereas overexpression of *TaCIPK14* resulted in enhanced wheat susceptibility to Pst by decreasing different aspects of the defense response, including accumulation of reactive oxygen species (ROS) and expression of pathogenesis-relative genes [19]. Currently, there is no information about CBL–CIPK participation in the regulation of biosynthesis of plant secondary metabolites.

In this study, we aimed for a comprehensive analysis of *CBL* and *CIPK* gene expression in the wild-growing grapevine *Vitis amurensis* Rupr. under the treatments inducing active production of stilbenes, which are important defense compounds in grapevines and some other plant families [20,21]. For this, we induced stilbene biosynthesis using stress hormones, ultraviolet C irradiation, and stilbene precursors to measure *CBL* and *CIPK* mRNA levels by quantitative real-time RT-PCR (qRT-PCR) in the leaves and callus cell culture of *V. amurensis*.

## 2. Results

### 2.1. Stilbene Content in V. Amurensis after SA, MeJA, CA, and UV-C Treatments

To test the involvement of *VaCBL* and *VaCIPK* genes in the regulation of stilbene synthesis in grapevine, we exposed vine cuttings and calli *V. amurensis* to different doses of SA, MeJA, stilbene precursors, and UV-C. We used experimental design and chemical concentrations for stress hormones (salicylic acid or SA, methyl jasmonate or MeJA), CA, and UV-C as described [22,23,24,25]. Previously, we found that the doses of the treatments (50–200 µM for SA and MeJA, 100–500 µM for CA, and 20 min at a wavelength of 254 nm, 230 µW cm^−2^, 15 cm from the UV-C lamp) exhibited the highest stimulating effect on stilbene biosynthesis. The highest stilbene content in the leaves of *V. amurensis* vines was observed 24 h after MeJA or UV-C treatments [26,27]. The highest stilbene content in callus cell cultures of *V. amurensis* was detected at 35 day of cultivation [22,28]. Therefore, we used the same time periods for stilbene analysis.

First, we analyzed the composition of stilbenes in the leaves and V7 callus cell culture of *V. amurensis*. The V7 callus cultures were established in 2017 from young stems of mature *V. amurensis* vines [29]. The HPLC analysis revealed the presence of eight major stilbenoids in the tissues of *V. amurensis* at a concentration of 0.001 mg/g DW or higher (Figure 1, Appendix A). It is possible that other stilbenes were present in the analyzed tissues of *V. amurensis*, but in trace amounts (less than 0.001 µg/g DW). The main focus of the present paper was to analyze the repertoire and content of the major stilbenes, and, therefore, we analyzed the content of the above-mentioned compounds.

The V7 callus cell culture of *V. amurensis* produced these eight stilbenes (Figure 1, Appendix A), among which five stilbenes (*t*-resveratrol diglucoside, *t*-piceid, *t*-resveratrol, *ε*-viniferin, and *δ*-viniferin) were present at consistently high levels (Figure 1, Appendix A). Moreover, *t*-resveratrol diglucoside (0.23–1.31 mg/g DW, 40.0–65.1% of all detected stilbenes) was specific to the calli of *V. amurensis* and was not found in the grapevine leaves [20]. However, cis-piceid, *t*-piceatannol, and *cis*-resveratrol were present in trace amounts in the calli of *V. amurensis*. Only UV-C irradiation considerably increased the total content of stilbenes to 0.2–0.3 mg/g DW (Appendix A).

The leaves of *V. amurensis* were characterized by the presence of seven major stilbenes (Figure 1, Appendix A), including *t*-piceid, *cis*-piceid, *t*-piceatannol, *t*-resveratrol, *cis*-resveratrol, *ε*-viniferin, and *δ*-viniferin. Among all detected stilbenes, five compounds (*t*-piceid, *cis*-piceid, *t*-resveratrol, *cis*-resveratrol, and *ε*-viniferin) were present at consistently high values (up to 0.16 mg/g DW), especially after CA or UV-C application (Figure 1, Appendix A). *t*-piceatannol and *δ*-viniferin were present at low levels, no more 0.01 mg/g DW, in all samples of grapevine leaves and callus cell culture (Appendix A).

All treatments considerably increased total stilbene content in both the cell culture and leaves of *V. amurensis* (Figure 2, Appendix A). In the calli of *V. amurensis*, the content of stilbenes increased most significantly after application of MeJA (200 µM) and UV-C (24 h post-treatment), up to 2.3–3.2 mg/g DW, which is 10.2- and 14.1-fold higher than in the untreated control calli, respectively (Figure 1a). For the leaves of *V. amurensis*, the highest stimulating effect on stilbene biosynthesis was provided by the application of SA (200 µM), CA (200 µM), and UV-C (24 h) with the increase in stilbene levels by 4.4, 5.4, and 5.5 times in comparison with the untreated control leaves, respectively (Figure 1b). We found that the UV-C treatment had the highest positive effect on stilbene biosynthesis compared with the effects of MeJA, SA, and CA, potentially due to the involvement of stilbenes in plant UV protection [20].

A detailed analysis of the individual stilbenes in the V7 probes (Appendix A) revealed that the significant increase in the total stilbene content after all treatments was mainly due to an increase in the content of viniferins (ε- and δ-viniferin), *t*-resveratrol, and *t*-resveratrol diglucoside (Appendix A), while in the leaves, the content of viniferins (ε- and δ-viniferin), *t*-resveratrol, and *t*-piceid was mostly enhanced after all used treatments (Table S1).

### 2.2. CBL and CIPK Expression after SA, MeJA, CA, and UV-C Treatments

Previously, eight *CBL* genes were described based on the well-known genome of cultivated grapevine *V. vinifera* [30]. However, in the NCBI GeneBank, two transcripts are associated with the *CBL4* gene of *V. vinifera*, four transcripts with the *CBL10a* gene, and two transcripts with the *CBL13* gene. These transcripts differed in the 3′-end of the *CBL* coding region and included modifications of the last exon (Appendix A), which resembled splicing-related variations. The specific primers were designed for all *CBL* transcripts found in the NCBI database (Appendix A). Thus, we analyzed the levels of 13 *VaCBL* transcript variants (Figure 3).

The qRT-PCR analysis revealed that the level of *VaCBL4-1* transcript considerably increased at all time points after UV-C treatment both in the V7 cell culture and grapevine leaves (Figure 3a,b). The levels of *VaCBL5*, *VaCBL10a-3*, and *VaCBL13-1* transcripts also considerably increased at both doses of SA (*VaCBL10a-3*) and MeJA (*VaCBL5*, *VaCBL10a-3*, *VaCBL13-1*), but this increase was detected only in cell cultures (Figure 3a). The levels of other *VaCBL* transcript variants were not significantly changed after the SA, MeJA, CA, and UV-C treatments of the grapevine *V. amurensis* cell cultures (Figure 3a). In contrast to the cell cultures, the treatments of *V. amurensis* vine cuttings led to a significant decrease in the levels of the most *VaCBL* transcripts (Figure 3b). Furthermore, expression of *VaCBL10a-1*, *10b*, and *13-3* transcripts increased in some individual probes (Figure 3). It is possible that *VaCBL4-1* is implicated in stilbene biosynthesis in grapevines as a positive regulator in response to UV-C irradiation. Transcripts *VaCBL10a-1*, *10b*, and *13-3*, *VaCBL5*, *VaCBL10a-3*, and *VaCBL13-1* may also have a slight positive effect on the biosynthesis of stilbenes, because their expression increased with only one dose of the used inductors (Figure 3).

The data obtained revealed that expression levels of five *VaCBLs* considerably decreased at both doses of the used stilbene biosynthesis inductors in the leaves of *V. amurensis*, including *VaCBL8* (UV-C), *VaCBL10a-2* (MeJA, CA), *VaCBL10a-4* (MeJA), *VaCBL11* (UV-C), and *VaCBL12* (MeJA) transcripts (Figure 3b). Furthermore, down-regulation of *CBL* transcript levels was detected at individual doses for the *VaCBL4-2*, *VaCBL5*, and *VaCBL10a-4* (Figure 3).

Then, we analyzed transcript levels of *VaCIPK* genes in response to SA, MeJA, CA, and UV-C (Figure 4). Previously, 20 *CIPK* genes were described for grapevines based on the well-known genome of cultivated grapevine *V. vinifera* [30]. However, in the NCBI GeneBank, two transcripts are associated with the *CIPK9* gene of *V. vinifera*, two transcripts—with the *CIPK21* gene, and six transcripts—with the *CIPK41* gene. These transcripts differ in the 3′-end of the *CBL* coding region and include modifications of the last exon (Appendix A). The specific primers were designed for all *CBL* transcripts found in the NCBI database (Appendix A). Thus, we analyzed the levels of 27 *VaCIPK* transcript variants (Figure 3).

Similarly, to *VaCBLs*, transcript levels of most *VaCIPKs* considerably decreased after the application of SA, MeJA, CA, and UV-C (Figure 4), which was expected due to the known CBL–CIPK interaction. Therefore, the list of calcium sensor genes negatively regulating stilbene accumulation also includes *VaCIPK3*, *9-1*, *9-2*, *12*, *21-1*, *21-2*, *33*, *34*, *35*, *36*, *37*, *39*, *40*, *41-3*, and *41-4*. However, there were several *VaCIPKs* with considerably elevated transcript levels in response to SA, MeJA, CA, and UV-C application, including *VaCIPK31*, *VaCIPK41-1*, and *VaCIPK41-2* (Figure 4a). Therefore, *VaCIPK31* (increased after MeJA treatment), *VaCIPK41-1* (increased after MeJA), and *VaCIPK41-2* (increased after CA) may play a role in the activation of stilbene accumulation in response to MeJA and CA. Furthermore, we detected the activation of *VaCIPK29*, *VaCIPK32*, *VaCIPK38*, *VaCIPK41-5*, and *VaCIPK41-6* transcript levels after application of only one dose of the CA, SA, UV-C, and MeJA treatments in grapevine cell cultures and leaves (Figure 4a).

## 3. Discussion

Previous data revealed the involvement of some *CDPK* and *CML* genes in the regulation of stilbene production in *V. amurensis* [29,31,32]. Five families of calcium sensor genes are known in plants; these are CDPK, CaM, CML, CBL, and CIPK [4,5,6]. Expression of *CDPK*, *CaM*, and *CML* was studied in grapevine tissues under stilbene biosynthesis-inducing conditions, including stress hormones (SA and MeJA), phenolic precursors (CA), UV-C, transformation by agrobacterium *rol* genes, leading to a strong increase in stilbene accumulation [29,31,32,33,34,35,36,37]. Thus, together with this study, the expression of main plant calcium sensors has been studied in grapevine cell cultures and leaves under conditions inducing stilbene biosynthesis.

It has been shown that the plant transformation with the *rol* genes of *Agrobacterium rhizogenes* was accompanied by a strong increase in *t*-resveratrol content and activation of the expression of *VaCDPK1d*, *VaCDPK1e*, *VaCDPK1-L*, and *VaCDPK2a* genes [34]. A number of *CML* genes, such as *VaCML52*, *VaCML65*, *VaCML93*, and *VaCML*95 [29], were highly up-regulated in the leaves and cell cultures of wild grapevine *V. amurensis* in the stilbene-modulating conditions. At the same time, *CaM* gene expression did not increase significantly [29].

A number of studies revealed that overexpression of the *AtCPK1* gene from *A. thaliana* contributes to a significant increase in the content of isoflavonoids in soybean [38]. Overexpression of *AeCDPK6* promoted hyperoside (flavonol, 3-O-galactoside of quercetin) accumulation in an *AeMYB30*-dependent manner in the flowers of okra plants [39]. The grapevine *V. amurensis* calli overexpressing the *VaCPK1*, *VaCPK26*, and *VaCPK20* genes were capable of producing up to 0.6, 0.4, and 4.2 mg/g DW of *t*-resveratrol, respectively [31,36]. However, overexpression of other *CDPK* genes in grapevine callus cultures did not significantly affect the biosynthesis of stilbenes [37]. For *CMLs*, it has been shown that overexpression of the *VaCML65* gene led to a considerable and consistent increase in the content of stilbenes of 3.8–23.7 times (up to 19 mg/g DW of total stilbenes or up to 18 mg/g DW of *t*-resveratrol) in all transformed lines [32].

In this work, the expression of *VaCBLs* and *VaCIPKs* has been analyzed after treatment with SA, MeJA, CA, and UV-C. Taken together, the data obtained in this study indicated that several *VaCBLs* (*VaCBL4-1*, *VaCBL10a-3*, *VaCBL13-1*) and *VaCIPKs* (*VaCIPK31*, *VaCIPK41-1*, *VaCIPK41-2*) can be considered as possible positive activators of stilbene biosynthesis, because their expression was considerably increased in after stilbene biosynthesis induction. The *VaCBL4-1* transcript is the most interesting candidate in the positive regulation of stilbene biosynthesis, because the *VaCBL4-1* transcript level markedly increased at all time points both in cell culture and grapevine leaves after UV-C treatment, which is the strongest inductor of stilbene biosynthesis (Figure 3). At the same time, the number of CBLs probably involved in negative regulation of stilbene accumulation is greater than the number of CBLs involved in the stimulation of stilbene accumulation. The results revealed that a high number of *VaCBL* (*VaCBL8*, *10a-2*, *10a-4*, *11*, *12*) and *VaCIPK* (*VaCIPK3*, *9-1*, *9-2*, *12*, *21-1*, *21-2*, *33*, *34*, *35*, *36*, *37*, *39*, *40*, *41-3*, *41-4)* family members may act as possible blockers of stilbene biosynthesis, since expression of these genes markedly decreased in the probes with high stilbene content. These genes are perhaps involved in grapevine primary metabolism and are responsible for growth and development regulation, since expression of these genes is suspended under the imposed stress caused by the stress hormones, stilbene precursors, and UV-C. In general, these findings need to be verified in future studies, potentially by establishing transgenic plants or plant cell cultures overexpressing the *CBLs* or *CIPKs* genes.

In conclusion, this study indicates the role of some CBL and CIPK family members in the regulation of secondary metabolite production in plant cells and provides new data on stilbene biosynthesis regulation in grapevines. According to this study, several *CBL* and *CIPK* genes can be used in biotechnology as stimulators for stilbene biosynthesis in plant cells.

## 4. Materials and Methods

### 4.1. V. amurensis Plant Material

We used young vines of wild-growing grapevine *V. amurensis* Rupr. (Vitaceae) sampled from a non-protected natural population near Vladivostok, Russia [40]. The *V. amurensis* vines collected in June 2022 were divided into cuttings (excised young stems 7–8 cm long with one healthy leaf) that were placed in individual 100 mL beakers with 50 mL filtered water and used for the treatments. The cuttings of *V. amurensis* were grown under a photoperiod of 11/13 h light/dark at a light intensity of 70 μmol m^−2^s^−1^ at 15 °C and darkness at 10 °C, humidity 60–70%.

The V7 callus cultures were established in 2017 from young stems of mature *V. amurensis* vines as described [29]. For stilbene analysis, the V7 calli were cultivated with 35-day subculture intervals in the dark at 24–25 °C in test tubes (height 150 mm, internal diameter 146 mm) with 7–8 mL of the solid Murashige and Skoog (MS) modified W_B/A_ medium [41,42] supplemented with 0.5 mg/L 6-benzylaminopurine, 2 mg/L α-naphthaleneacetic acid, and 8 g/L agar.

### 4.2. Plant Treatments and Chemicals

Sterile solutions of SA diluted in ethyl alcohol, MeJA and CA diluted in dimethyl sulfoxide (DMSO) were added aseptically to desired concentrations (50 and 200 µM of SA and MeJA, 100 and 500 µM of CA) to the autoclaved cultivated media for cell cultures or to the filtered water for the vine cuttings of *V. amurensis*. DMSO (200 µL/L) or ethyl alcohol (200 µL/L) were added for the control treatments (V7 or L). The UV-C treatments were conducted for 20 min at a wavelength of 254 nm (230 µW cm^−2^, 15 cm from the lamp) in the dark using UV lamp VL-215.MC provided by Vilber Lourmat company (Marne-la-Vallée, France) as described [24]. Control *V. amurensis* cuttings were placed in the dark in individual 100 mL beakers with 50 mL filtered water. After the UV-C treatments, the treated and control cuttings were grown under a photoperiod of 11/13-h light (70 μmol m^−2^s^−1^, 15 °C)/dark (10 °C) for 1 h and 24 h.

### 4.3. Stilbene Analysis by High Performance Liquid Chromatography (HPLC) and Mass Spectrometry

Total stilbene content was measured by HPLC as described [43,44]. Briefly, the extracts were separated on Zorbax C18 column (150 mm, 2.1-mm i.d., 3.5-lm part size, Agilent Technologies, USA). The mobile phase consisted of a gradient elution of 0.1% aqueous acetic acid (A) and acetonitrile (B). The gradient profile with a flow rate of 0.2 mL/min was: 0 min 0% B; 35 min 40% B; 40 min 50% B; 50 min 100% B; and then eluent B for 60 min. Mass spectrometry was performed using a 1260 Infinity analytical system (Agilent Technologies, Santa Clara, CA, USA) for quantification of all components as described [43]. HPLC was performed using LC-20 analytical HPLC system (Shimadzu, Japan), equipped with an SPD-M20A photodiode array detector, LC-20ADXR pump, Shim-pack XR-ODS II column and SIL-20ACXR auto sampler for quantification of all components as described [44].

### 4.4. RNA Isolation, Reverse Transcription, and Expression Analysis of VaCBLs and VaCIPKs

Total RNA extraction was performed using the cetyltrimethylammonium bromide (CTAB)-based extraction as described [33]. Complementary DNAs were synthesized using 1.5 µg of RNA by the MMLV Reverse Transcription PCR Kit with oligo(dT)15 (RT-PCR, Evrogen, Moscow, Russia). The RT-PCR reactions were performed in 50 µL aliquots of the reaction mixture, which contained 1 × RT buffer, 1 mM each of the 4 dNTPs, 2 µM of dithiothreitol (DTT), 1 µM of oligo(dT)15 primer, 250 U of MMLV-polymerase at 37 °C for 1.5 h. The 1 µL samples of reverse transcription products were then amplified by qPCR.

qRT-PCRs were performed using a real-time PCR kit (Evrogen, Moscow, Russia) and SYBR Green I for qRT-PCR (Evrogen, Russia) using cDNAs as described [40,44]. The expression was calculated by the 2^−ΔΔCT^ method [45] with two internal controls (*VaGAPDH* and *VaActin1*) in all experiments as described [40]. qRT-PCR data were obtained from free independent experiments and are averages of 12 technical replicates for each independent experiment (6 qPCR reactions normalized to *VaGAPDH* and 6 qPCR reactions normalized to *VaActin1* expression). Expression data are presented as a heatmap map in the program ComplexHeatmap [46]. The specific primers were designed for all *CBL* and *CIPK* transcripts of *V. vinifera* found in the NCBI database (Appendix A) according to the previously published paper [30].

## Figures and Tables

**Figure 1 plants-12-01562-f001:**
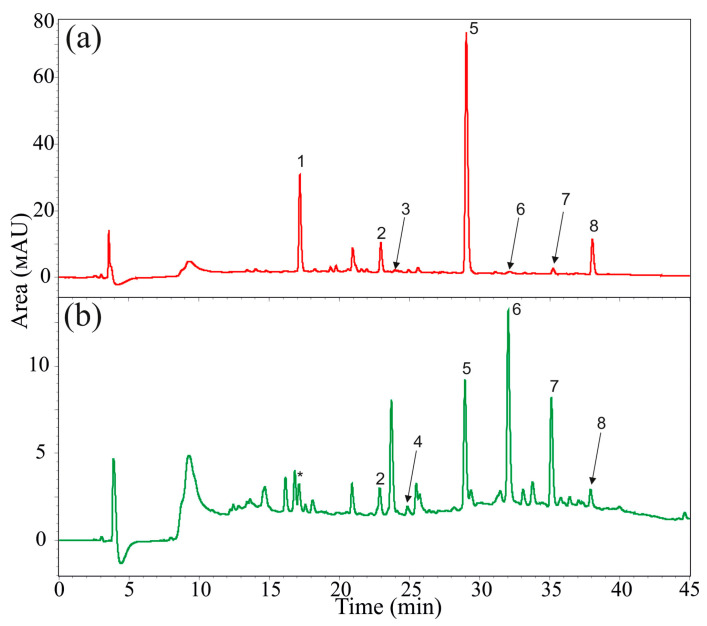
A representative HPLC-UV profile for the extracts obtained from *Vitis amurensis* and recorded at 310 nm. (**a**) V7 callus cell culture of *V. amurensis*; (**b**) leaves of *V. amurensis* after ultraviolet C irradiation. *t*-resveratrol diglucoside (1), *t*-piceid or *t*-resveratrol glucoside (2), *cis*-piceid or *cis*-resveratrol glucoside (3), *t*-piceatannol (4), *t*-resveratrol (5), *cis*-resveratrol (6), ε-viniferin (7), δ-viniferin (8). (*) The compound was similar in retention time to *t*-resveratrol diglucoside, but showed other parameters of UV absorption spectrum (310 nm).

**Figure 2 plants-12-01562-f002:**
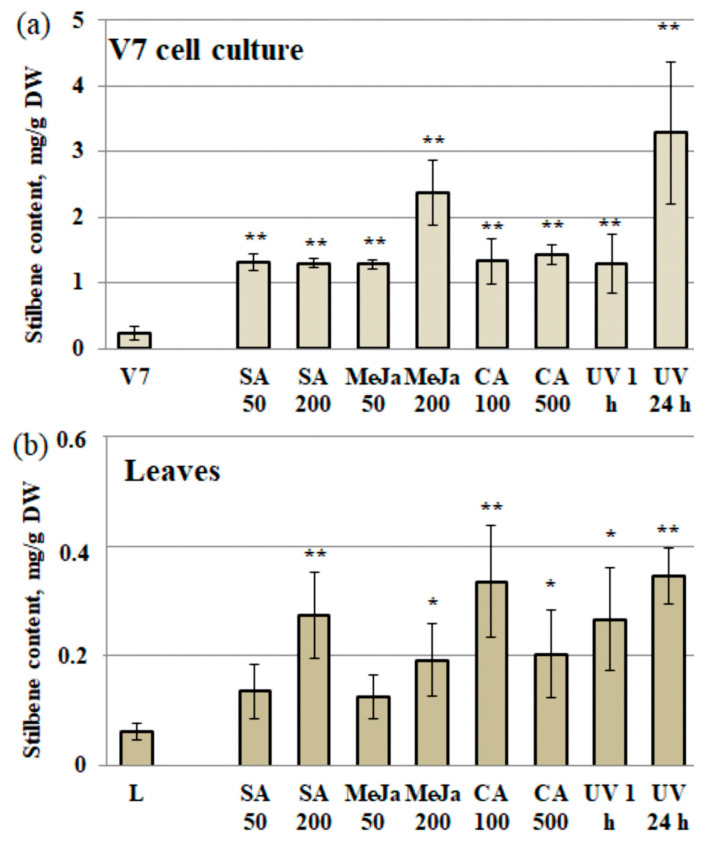
Total stilbene content in the cell culture and leaves of *Vitis amurensis* after treatments with salicylic acid (SA), methyl jasmonate (MeJA 50, MeJA 200 µM), *p*-coumaric acid (CA 0.1, CA 0.5 mM) and ultraviolet C irradiation (UV 1 h, UV 24 h). (**a**) V7 callus cell culture of *V. amurensis*; (**b**) leaves of *V. amurensis*. L or V7—control conditions; SA 50 and SA 200—salicylic acid 50 and 200 µM; MeJA 50 and 200—methyl jasmonate 50 and 200 µM; CA 100 and 500—*p*-coumaric acid 100 and 500 µM; UV 1 h and UV 24 h—ultraviolet C irradiation. The leaves of *V. amurensis* samples were harvested 24 h after SA, MeJA, or CA application and 1 h/24 h after UV-C irradiation. The calli samples were harvested from the 35-day-old cell culture V7 of *V. amurensis* growing in in the presence of SA, MeJA, or CA or 1 h/24 h after UV-C irradiation. *—*p* < 0.05; **—*p* < 0.01 versus values of total stilbene levels in the V7 cell line or in grape leaves cultivated under control conditions.

**Figure 3 plants-12-01562-f003:**
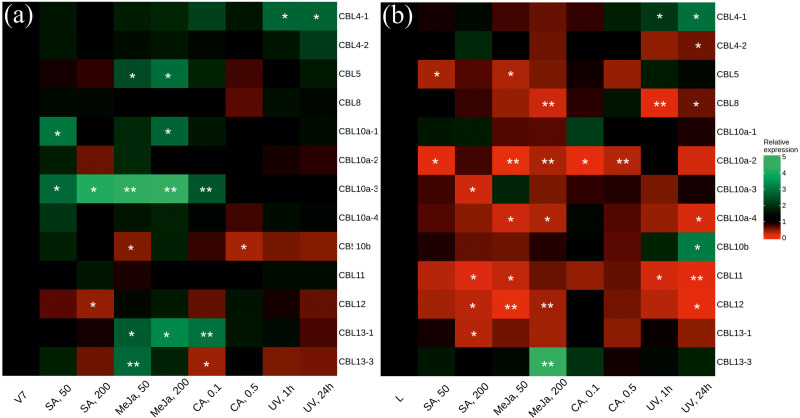
Heatmap of *VaCBL* expression levels in *Vitis amurensis* after salicylic acid (SA), methyl jasmonate (MeJA), *p-*coumaric acid (CA) and ultraviolet irradiation (UV-C) treatments. (**a**) V7 callus cell culture of *V. amurensis*; (**b**) leaves of *V. amurensis*. The *VaCBL* transcript levels were determined by quantitative RT-PCR. The color scale represents increased (green) and decreased (red) changes of the expression values under the SA, MeJA, CA, and UV-C treatments relative to the control. V7 and L—control non-treated conditions (for V7 cell cultures 35 d growth in W_B/A_ medium; for leaves—24 h in filtered water at 25 °C); SA 50 and SA 200—supplemented with 50 and 200 µM of SA; MeJA 50 and MeJA 200—supplemented with 50 and 200 µM of MeJA; CA 0.1 and 0.5—supplemented with 100 and 500 µM of CA; UV 1 h and 24 h—1 h or 24 h after UV-C irradiation. *, **—significantly different from the values of *VaCBL* expression in *V. amurensis* cells under the control conditions (V7 or L) at *p* ≤ 0.05 and 0.01 according to the Student’s *t*-test.

**Figure 4 plants-12-01562-f004:**
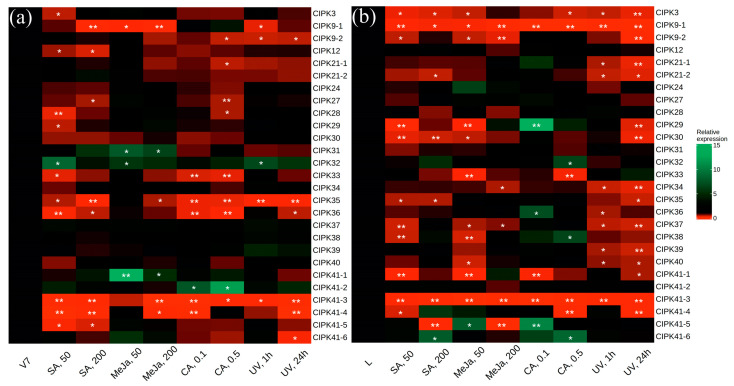
Heatmap of *VaCIPK* expression levels in *Vitis amurensis* after salicylic acid (SA), methyl jasmonate (MeJA), *p-*coumaric acid (CA) and ultraviolet irradiation (UV-C) treatments. (**a**) V7 callus cell culture of *V. amurensis*; (**b**) leaves of *V. amurensis.* The *VaCBL* transcript levels were determined by quantitative RT-PCR. The color scale represents increased (green) and decreased (red) changes of the expression values under the SA, MeJA, CA, and UV-C treatments relative to the control. V7 and L—control non-treated conditions (for V7 cell cultures 35 d growth in W_B/A_ medium; for leaves—24 h in filtered water at 25 °C); SA 50 and SA 200—supplemented with 50 and 200 µM of SA; MeJA 50 and MeJA 200—supplemented with 50 and 200 µM of MeJA; CA 0.1 and 0.5—supplemented with 100 and 500 µM of CA; UV 1 h and 24 h—1 h or 24 h after UV-C irradiation. *, **—significantly different from the values of *VaCIPK* expression in *V. amurensis* cells under the control conditions (V7 or L) at *p* ≤ 0.05 and 0.01 according to the Student’s *t*-test.

## Data Availability

The data presented in this study are available within the article and Appendix A.

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
