# Peer review of "The Effect of Stress Hormones, Ultraviolet C, and Stilbene Precursors on Expression of Calcineurin B-like Protein (CBL) and CBL-Interacting Protein Kinase (CIPK) Genes in Cell Cultures and Leaves of Vitis amurensis Rupr"

_plants, 2023, doi:10.3390/plants12071562_

Round 1
Reviewer 1 Report
Kiselev and his colleagues aim to investigate expression of calcineurin B-like proteins (CBL) and CBL-interacting protein kinase (CIPK) genes in response to conditions inducting stilbenes biosynthesis in grapevine. The manuscript provides new information about the relationship between expressions of CBL and CIPK gene and the stilbene contents in V. amurensis after SA, MeJA, CA, and UV-C treatments. The conclusions are supported by the data, and the submitted manuscript is written clearly and general interest to the readers. However, I have several comments that should be addressed before publication.
In scientific aspects, I have some comments:
1. In the Materials and Methods part, the author should tell us how obtain the Nucleotide sequences of the VaCBLs and VaCIPKs genes tested in the manuscript (Page9, L339-340).
2. Two internal controls (VaGAPDH and VaActin1) (Page9, L337-338) selected for qRT-PCRs, the author should give more details. If the all the qRT-PCRs were performed with both internal controls, or some time used the VaGAPDH, the remains used the VaActin1.
3. The authors suggested that CBL-like transcripts differed in the 3ꞌ-end of the coding region and included modifications of the last exon in V. vinifera. If the sequences changes correlated with transcripts variation among different VaCBLs, or the sequences changes result in transcripts variation? There will be discussed in the discussion part.
In language aspect:
1. Protein names should be in Roman format (Page1, L18). Please check the whole manuscript。
2. The abbreviation of Probability in“p ≤ 0.05” should be in italic format(Page 6, L200).
Author Response
Reviewer 1
We are grateful to the Reviewer for the careful evaluation of our manuscript. We provided detailed answers to the Reviewer’s comments below.
1) “1. In the Materials and Methods part, the author should tell us how obtain the Nucleotide sequences of the VaCBLs and VaCIPKs genes tested in the manuscript (Page 9, L339-340).”
- Answer: Thanks for the comment. Indeed, we forgot to include these details in Materials and Methods. In the revised manuscript (Ms), we included this information in the Material and Methods: “The specific primers were designed for all found CBL and CIPK transcripts of V. vinifera found in the NCBI database (Table S2) according the previous published paper [30].” Please, see Line 332-334.
2) “2. Two internal controls (VaGAPDH and VaActin1) (Page9, L337-338) selected for qRT-PCRs, the author should give more details. If the all the qRT-PCRs were performed with both internal controls, or some time used the VaGAPDH, the remains used the VaActin1.”
- Answer: In all experiments, we used two internal controls (both VaGAPDH and VaActin1). To address this comment, we included in revised Ms text detailed information: “The expression was calculated by the 2−ΔΔCT method [37] with two internal controls (VaGAPDH and VaActin1) in all experiments as described [31]. qRT-PCR data were obtained from free independent experiments and are averages of 12 technical replicates for each independent experiment (6 qPCR reactions normalized to VaGAPDH and 6 qPCR reactions normalized to VaActin1 expression).”.
Please, see line 327-331.
3) “3. The authors suggested that CBL-like transcripts differed in the 3ꞌ-end of the coding region and included modifications of the last exon in V. vinifera. If the sequences changes correlated with transcripts variation among different VaCBLs, or the sequences changes result in transcripts variation? There will be discussed in the discussion part.”
- Answer: According to GeneBank data, we suggested that these transcripts were transcript variations due to alternative splicing. We included this information in the revised Ms text. Please, see Line 168.
4) “In language aspect:
1.Protein names should be in Roman format (Page1, L18). Please check the whole manuscript.”
- Answer: Corrected.
5) “2.The abbreviation of Probability in“p ≤ 0.05” should be in italic format (Page 6, L200).”
- Answer: Corrected.
Reviewer 2 Report
The ms was aimed to study the effect of stress hormones, ultraviolet C, and stilbene pre-2 cursors on the expression of calcineurin B-like protein (CBL) and 3 CBL-interacting protein kinase (CIPK) genes in cell cultures 4 and leaves of Vitis amurensis Rupr. under the treatments inducing active production of stilbenes, which are important defense compounds in grapevine and 86 some other plant families
It is well-written and understandable and very clear. The conclusion consists of the evidence and argument discussed well.
The topic studied is original. They obtained new findings on CBL and CIPK 32 implications in the regulation of plant secondary metabolism in response to stress hormones, metabolite precursors, and UV-C irradiation.
The only issue is that the last sentence of the introduction section should be merged with the previous sentence which is describing the aim of the study or it would be better to start with: "For this, we induce..."
Author Response
Reviewer 2
6) ‘The only issue is that the last sentence of the introduction section should be merged with the previous sentence which is describing the aim of the study or it would be better to start with: "For this, we induce..."”
- Answer: Corrected.